# A Direct Observation Video Method for Describing COVID-19 Transmission Factors on a Micro-Geographical Scale: Viral Transmission (VT)-Scan

**DOI:** 10.3390/ijerph18179329

**Published:** 2021-09-03

**Authors:** Richard R. Suminski, Gregory M. Dominick, Norman J. Wagner

**Affiliations:** 1Center for Innovative Health Research, Department of Behavioral Health and Nutrition, University of Delaware, Newark, DE 19726, USA; 2Department of Behavioral Health and Nutrition, University of Delaware, Newark, DE 19726, USA; gdominic@udel.edu; 3Department of Chemical & Biomolecular Engineering, University of Delaware, Newark, DE 19716, USA; wagnernj@udel.edu

**Keywords:** observation method, pandemic, infectious disease, public health, health behavior, measurement

## Abstract

The COVID-19 pandemic severely affected many aspects of human life. While most health agencies agree mask wearing and physical distancing reduce viral transmission, efforts to improve the assessment of these behaviors are lacking. This study aimed to develop a direct observation video method [Viral Transmission (VT)-Scan] for assessing COVID-19 transmission behaviors and related factors (e.g., environmental setting). A wearable video device (WVD) was used to obtain videos of outdoor, public areas. The videos were examined to extract relevant information. All outcomes displayed good to excellent intra- and inter-reliability with intra-class correlation coefficients ranging from 0.836 to 0.997. The majority of people had a mask (60.8%) but 22.1% of them wore it improperly, 45.4% were not physical distancing, and 27.6% were simultaneously mask and physical distancing non-compliant. Transmission behaviors varied by demographics with white, obese males least likely to be mask-compliant and white, obese females least likely to physical distance. Certain environments (e.g., crosswalks) were identified as “hot spots” where higher rates of adverse transmission behaviors occurred. This study introduces a reliable method for obtaining objective data on COVID-19 transmission behaviors and related factors which may be useful for agent-based modeling and policy formation.

## 1. Introduction

The SARS-CoV-2 virus that causes coronavirus disease 2019 (COVID-19), became a global public health crisis in early 2020. Since then, the pandemic has grown considerably not only in reach but in magnitude (e.g., number of cases per week). In just over a year and a half, there have been nearly 212 million cases globally with over 4.4 million virus-related deaths. In the U.S. alone, 37.4 million cases have been confirmed with 622,459 deaths as of 23 August 2021 [1].

COVID-19 is caused by an airborne pathogen that spreads by inhaling or ingesting respiratory droplets and aerosols that are expelled into the air from an infected human carrier. The virus is highly contagious and recent studies have reported that it has a higher transmission rate than the seasonal influenza virus. Moreover, it is now known that carriers can be both symptomatic and asymptomatic [2,3,4]. In the absence of effective antiviral treatments and vaccines, many governments have adopted various non-pharmaceutical interventions (i.e., COVID-19 transmission behaviors) in an attempt to slow the pandemic. The primary behaviors include physical distancing (maintaining a physical distance of at least six feet from people not from your household in both indoor and outdoor spaces), use of a face covering or mask, and touching fomites (objects or materials which are likely to carry infection) and then touching areas around the face (e.g., mouth, nose, or eyes) [5]. It is clear that physical distancing is an effective approach and most recent evidence indicates mask wearing, when done correctly, also mitigates the spread of COVID-19 viral particles [6,7,8]. It should be pointed out, however, that while masks are efficacious their effectiveness is questionable mainly due to adherence resistance resulting from a myriad of reasons such as the politicization of mask use [9,10]. Other factors appear to enhance/inhibit the effectiveness of transmission prevention behaviors. These include personal (e.g., racial background, age, weight status) and environmental (e.g., temperature) characteristics [11].

An important aspect of public health research pertains to the measurement of human behavior with particular relevance to the inter-relationship between feasibility and reliability/validity [12,13]. Often the most feasible (i.e., lowest cost) assessment methods of human behaviors contain the most bias or error—they are less accurate in their description of the behavior. For instance, calories expended while engaged in physical activity behaviors can be measured with a whole gamut of tools ranging from the most feasible, least accurate (e.g., self-report surveys) to the least feasible, most accurate (e.g., doubly labeled water). Practically all research (bench to practice) on COVID-19 including prevention and mitigation efforts, relies at some point on the measurement of COVID-19 transmission behaviors and factors. Indeed, classic deterministic computational models such as Susceptible-Infected-Recovered (SIR) and its variants especially the more specific, finer-grained, agent-based models, tend to rely heavily on transmission behavior assessments to project the progression of an infectious disease and likely epidemic outcomes [14,15].

Even with an exponential rise in COVID-19 research, important transmission behaviors are still commonly evaluated using self-report methods (e.g., surveys) with a few studies analyzing surveillance videos mainly to detect mask utilization [16,17,18,19]. While the former is more feasible and can provide information on large samples of people at the individual-level, the later possesses the advantages afforded by direct observation—no recall or conformity biases, high accuracy rates at geographical or behavior setting (e.g., airport) levels [13]. However, the use of surveillance video for public health purposes has some drawbacks that could partially or fully diminish the relevance of its use for assessing COVID-19 transmission. Permission to access surveillance video is almost always needed and obtaining permission can be a cumbersome, time-consuming process that does not consistently result in an affirmative response. Further, issues related to camera placement, quality, maintenance, and functionality can be troublesome especially when focusing on something other than public safety and crime prevention—the most common purposes for surveillance cameras [20]. Direct observation also has been used to describe mask wearing and physical distancing [21,22,23,24]. Although this approach overcomes some of the disadvantages mentioned above, the studies that used it either provided limited or no information regarding its reliability, restricted observations to a few, specific behavior settings (e.g., grocery store), and/or lacked details about the method [21,22,23,24]. Further, they employed observation methodology that has been shown to be inaccurate when describing larger groups of people [21,22,23,24].

To address some of these limitations, the current study developed an observation protocol that uses a high-resolution, wearable video device (WVD) to capture data on COVID-19 transmission behaviors and related factors across a wide range of environmental settings. The use of video has been shown to improve the accuracy of observations when studying physical activity behavior [25,26]. In addition, this study provides a comprehensive set of rules for extracting data from video that is vital for scientific rigor and reproducibility. Lastly, both inter- and intra-rater reliability are examined for all outcomes including the geolocation (longitude and latitude coordinates) of those people described. We present outcomes for the reliability of VT-Scan and apply VT-scan in a case study.

## 2. Materials and Methods

### 2.1. Overview

This study was comprised of two phases. The first was a pilot test to obtain reliability metrics on VT-scan. The second was a case study demonstrating some applications of VT-scan. Data for the pilot test were collected the week prior to data collection for the case study. The case study utilized data collected over a six-week period with the first week (baseline) occurring immediately before the start of the fall semester. Details for each phase are provided below for the various study aspects.

### 2.2. Data Collection Device

The Gogloo E7 SMART is a state-of-the-art WVD indistinguishable from a pair of normal sunglasses (Figure 1). The camera is discretely centered in the bridge of the glasses which provides (Gogloo E7 SMART Eyewear, Model Number E7B0100 [27]).

Video capture from the user’s point-of-view. It features an 8 MP Sony Complementary Metal Oxide Semiconductor sensor for recording full 1080p high definition MP4 video at 30 frames per second. The glasses accept a 64 Gb memory card allowing up to 8 h of recording at 1080p, have a self-contained, lithium polymer battery providing six to eight hours of recording time, a 110-degree field of vision which approximates the human 90 degree field of vision, and video files can be saved with time/date/GeoCoordinate stamps.

### 2.3. Observation Route (Setting)

Videos were collected along a 4.69 mi pre-determined route that coursed through the University of Delaware campus (60% of route) and adjacent business and residential areas (40% of route) (Figure 2). The route was constructed to connect 79 nodes which represented areas normally associated with high volumes of pedestrian traffic [entrances/exits of academic buildings, entrances/exits of dormitories, dining halls, recreational and student facilities (Figure 3A); campus walkway pinch points (area where walkway is narrowed by obstruction) (Figure 3B); sidewalks; crosswalks (Figure 3C) and public open spaces (Figure 3D)]. The route between notes consisted of campus walkways and residential and business area sidewalks/streets.

### 2.4. Observation Procedure

All videos were collected by one investigator (RRS—observer) over a seven-week period on one weekday per week between 17 August and 28 September 2020. The weekday for pilot and baseline data collection was randomly selected (Monday was selected) and occurred prior to the start of the fall semester. The weekdays for the remaining data collections days were Tuesday (week 3) through Friday and Monday (week 7). Observations for all data collection days started from the same beginning point at 11:15 am and ended at the same stop point between 12:15 and 12:19 pm. The observer traversed the observation route on a bicycle while recording videos with the WVD. Videos were obtained for 10 s at each node from the same position each observation day with the observer motionless. Videos also were captured continuously by the observer while moving from one node to another traveling only on campus walkways and residential/business area streets. Video footage obtained during inter-node travel contained the entire breadth of the campus walkways or streets including the sidewalks on either side. For people orientated in the same direction as the observer, the observer turned his head in order to obtain video of their face. Otherwise, the observer kept his head, and thus camera, focused on the relevant area of the inter-node route. These procedures helped video reviewers limit descriptions of people to specific route areas. See Figure 4 for images showing inter-node, street/sidewalk view with two people orientated in the same direction as observer (upper left of Figure 4A) and the same inter-node, street/sidewalk (Figure 4B) with side video capture of the facial areas of the two people seen in Figure 4A (one with mask on forehead and other with mask around neck). No precipitation fell on any of the observation days, the temperature (77.7 ± 6.3, range 69.0–87.0 °F), humidity (65.1 ± 4.8, range 57.0–71.0%) and wind speed (8.6 ± 2.8, range 5.0–12.0 mph) were typical for the area during the times data were collected [28].

### 2.5. Data Extraction

Reviewers were trained on the procedures and rules for extracting relevant data captured in the WVD videos. Training included practice sessions and further instruction with feedback on any errors. For deriving inter-rater reliability metrics, reviewer 1 and reviewer 2 independently examined the same ~60 min pilot video during review session 1 and for intra-rater reliability metrics, reviewer 1 examined the same pilot video one week later during review session 2. For the case study, videos collected at baseline and each of the five weeks after baseline were randomly assigned to a group of trained reviewers who independently assessed the videos.

Information was extracted from the videos using a number of computer-aided manipulations (i.e., zooming, slow-motion, rewind, pause) and a comprehensive set of rules (Table 1). The COVID-19 transmission behaviors were related to face mask compliance (have a mask, have mask but wearing it incorrectly, no mask or have a mask but wearing it incorrectly), physical distancing compliance (<6 ft vs. >6 ft from other people), interactions with fomites, and face touching. The personal characteristics were type of activity not moving (sitting or standing) and moving (walking, running, biking, skateboarding], sex/gender (male, female), age group (<18, 19–30, 31 to 55 and >55 years), race (white, non-white) using a 7-point scale related to skin tone (very light skin color—very dark skin color), and weight status using pictorial scales depicting not obese [body mass index (BMI) < 30] and obese (BMI ≥ 30) individuals [29,30,31]. Environmental factors were geographical location (longitude and latitude coordinates obtained using the Google map location tool) and node type (campus walkway, crosswalk, etc.). Timestamps (h:min:s) shown on the videos were used to mark when an individual was described.

### 2.6. Data Management and Statistical Analysis

A protocol for handling the videos was developed to mitigate ethical issues arising from collecting, storing and analyzing detailed video data. After each data collection period, the WVD’s micro secure digital (SD) card with acquired videos was removed and immediately transferred to a lock box for storage until transported to the computer lab. At the lab, the video data on the SD card was uploaded to a secure server and encrypted after which the SD card’s video files were erased. The encrypted video files were then distributed to reviewers who performed the reviews. Once reviews were complete, all videos in the possession of the reviewers were erased. The University’s Institutional Review Board for Protection of Human Subjects Committee waived the requirement for written informed consent for participants in this study under exemption #2: Research involving observation of public behavior, in accordance with the national legislation and the institutional requirements [32]. 

For the pilot study, intraclass correlation coefficients (ICCs) were calculated using the total number of people described by a reviewer per five min epoch (*n* = 12 epochs from the 60 min pilot video) as the unit of analysis and including only the category of an outcome with the largest sample size. Inter-rater reliabilities were assessed using ICCs estimated from single rater, consistency, 2-way random effects models while intra-rater reliability was assessed using ICCs estimated from single rater, absolute agreement, 2-way mixed effects models [33]. The ICCs were interpreted as follows: <0.5 = poor reliability, 0.5 to 0.75 = moderate reliability, 0.75 to 0.9 = good reliability, and >0.90 = excellent reliability [34]. Kappa statistics (*κ*) were derived to assess intra- and inter-rater agreement on the presence/absence of outcome variables. The level of agreement for the *κ* statistics were interpreted as follows: 0–0.20 none; 0.21–0.39 minimal; 0.40–0.59 weak; 0.60–0.79 moderate; 0.80–0.90 strong; >0.90 almost perfect [35]. Intra- and inter-rater differences in longitude/latitude coordinates obtained were examined using independent t-tests. For Kappa and *t*-test analyses, only the same individuals described by reviewer 1 during review sessions 1 and 2 (intra) and by reviewers 1 and 2 during review session 1 (inter) were included. To be considered the “same” individual, video time stamps recorded by the reviewers had to match to within ± 1 s. 

For the case study, Chi-square tests for independence with Yate’s continuity correction (for 2 × 2 tables only) were done to determine if transmission behaviors varied by demographics and geographical locations (crosswalks, non-academic and academic campus building exits/entrances, open spaces, campus walkways, and sidewalks). For the age group variable, individuals described as being under 18 years of age were removed from the Chi-square analysis due to the small sample size and no one in this age group practicing physical distancing. All statistical analyses were performed using the IBM Corp., SPSS, Armonk, NY, USA statistical software package with alpha set a priori at 0.05 [36].

## 3. Results

### 3.1. Pilot Study

Counts and ICCs for the five min epochs are provided in Table 2. All outcomes displayed good to excellent intra- and inter-reliability with ICCs ranging from 0.836 for age group (inter-rater) to 0.997 for surface touching (intra-rater). Eighty one percent of the intra-rater and 72.7% of the inter-rater ICCs were over 0.9 which is excellent agreement. The numbers of individuals described in the 60 min pilot video were similar across reviewers. Reviewer 1 described 126 people during review session 1 and 129 during review session 2 while reviewer 2 described 131 during review session 1. Numbers and percentages of individuals described for each outcome of interest were very similar between reviews with the largest spread for weight status estimates in review 1 [reviewer 1 *n* = 95 (75.4% of total) and reviewer 2 *n* = 111 (84.7% percent of total) not obese]. Out of the total numbers of individuals described, 111 individuals were described by reviewer 1 during review sessions 1 and 2 and 117 were described by both reviewer 1 and 2 during review session 1. All told, about 10% of the people described during each review session could not be timestamp matched to a person described during another corresponding review session.

Based on descriptions of the same people, *κ* statistics were calculated to assess intra- and inter-rater agreement on the presence/absence of characteristics. As can be seen in Table 3, all *κ* statistics were significant (*p* < 0.001) and according to the interpretation criteria, 60% of the intra-reviewer *κ* statistics indicated strong agreement while 40% indicated moderate agreement. For inter-reviewer, 40% of *κ* statistics were strong and 60% moderate. Intra-reviewer percent agreement averages were 92.8% and 91.8% for the five COVID-19 behaviors and personal characteristics, respectively. Average percent agreement for inter-reviewer COVID-19 behaviors was 93.6% and 89.1% for personal characteristics.

Intra-reviewer (3.61 ± 3.27 ft; *n* = 111) and inter-reviewer (3.04 ± 2.92 ft; *n* = 117) coordinate differences did not differ significantly (*p*
= 0.51). Intra- and inter-reliability coordinate differences did not differ significantly between people not moving vs. moving (*p*
= 0.17 for intra;
*p*
= 0.14 for inter), physical distancing vs. not physical distancing (*p*
= 0.31 for intra;
*p*
= 0.32 for inter), and at a node vs. not at a node (*p*
= 0.52 for intra;
*p*
= 0.18 for inter) (Table 4).

### 3.2. Case Study

A total of 916 people were described during the case study in 372 min of video which equates to 2.46 people/min. The fewest people were seen during the first week of data collection (*n* = 87), which was the week immediately prior to the start of the semester. With the onset of the semester, the numbers described increased substantially and remained fairly consistent throughout the remaining weeks of data collection (M = 165.8, SD = 27.6, range 127 to 200). 

Characteristics and COVID-19 transmission behaviors of those observed are provided in Table 5. Of note are the high percentages of the 916 people observed for whom personal and behavioral information could be ascertained from the videos. For instance, gender was discernable for 99% of the total observed and the presence/absence of a mask for 97.6%. The percentages in each personal characteristic category are very consistent with what would be expected from demographics for the area studied (men and women who are mostly white, college-aged, non-smokers with about 1 in 5 obese). More specifically, U.S. Census and national survey data corresponding to the study area indicates 56.9% were female, 83.2% were white, 78.1% were 18–34 years of age, and 20.8% were obese compared to the case study where 54.6% were female, 81.1% were white, 72.5% were 18–34 years of age, and 19.3% were obese [37]. In terms of COVID-19 transmission behaviors, the majority had a mask with them (60.9%), but 22.1% were not using the mask correctly (around neck, on forehead, in hand). This mask behavior was most likely to lead to the exposure of both the mouth and nose. Physical distancing was not practiced by 45.4% of those described while 27.6% were both mask and physical distancing non-compliant. Touching ones face with ones hands and touching fomites were also observed, but at much lower frequencies than the other transmission behaviors (6.7% and 18.5% for face and fomite touching, respectively). Further, when touching fomites, most people touched a cell phone (40.8%).

A breakdown of COVID-19 transmission behaviors by personal characteristics are provide in Table 6. No significant differences in behaviors were found between age groups. Mask non-compliance was more likely to be observed in whites than non-whites [*Χ*^2^ = (1, N = 1873) = 32.1; *p* < 0.001)] and those not moving compared to those moving [*Χ*^2^ (1, N = 1885) = 21.4; *p* < 0.001]. Non-compliance with physical distancing recommendations also varied by race and activity as well as sex/gender and weight status. Less compliance was seen in whites [*Χ*^2^ = (1, N = 1897) = 5.0; *p* = 0.03], females [*Χ*^2^ = (1, N = 1910) = 23.4; *p* < 0.001], obese [*Χ*^2^ = (1, N = 1901) = 6.6; *p* = 0.01], and individuals not moving [*Χ*^2^ = (1, N = 1914) = 7.4; *p* = 0.01]. Similar differences were noted (except for weight status) for simultaneous non-compliance with mask and physical distance recommendations for race [whites less compliant *Χ*^2^ = (1, N = 1873) = 19.2; *p* < 0.001], gender [females less compliant *Χ*^2^ = (1, N = 1878) = 5.8; *p* = 0.02], and activity [individuals not moving less compliant *Χ*^2^ = (1, N = 1882) = 22.7; *p* < 0.001]. The only significant difference in touching behavior was between individuals not moving and those moving. Those observed not moving were much more likely to touch a surface than those moving [42.6% vs. 9.5%; *Χ*^2^ = (1, N = 1903) = 129.3; *p* < 0.001].

### 3.3. Temporal and Spatial Descriptive Examination

In Figure 5, Figure 6 and Figure 7, visual representations are provided to show mask and physical distancing compliance across the pilot study weeks. Mask compliance varied as a function of study week because of the significant increase in compliance noted at week 6 after more stringent rules regarding COVID-19 transmission behaviors were imposed by the University [*Χ*^2^ (5, N = 5885) = 14.8; *p* = 0.01; adjusted standardized residual week 6 = 3.8] (Figure 5). The percentage of individuals mask compliant was very steady from baseline through week 5 at about 45% but increased substantially by 15% to 60.3% compliant in week 6. Physical distancing also varied by week; however, the pattern of variation was somewhat different than seen for mask compliance [*Χ*^2^ (5, N = 5914) = 19.0; *p* = 0.002] (Figure 6). At baseline, physical distancing compliance was highest (75.0%) but then dropped nearly 30% at weeks 2 and 3 followed by a 13% rebound at week 4 which was maintained through week 6. Lastly, in Figure 7 are the percentages of those observed simultaneously compliant with both mask wearing and physical distancing across study weeks. The combination of behaviors also varied by study week with the pattern essentially the product of the two individual behaviors [Χ^2^ (5, N = 5882) = 17.4; *p* = 0.002]. Mask and physical distancing compliance were high at baseline, dropped around 20% from weeks 2 through 4, then showed a rebound to near baseline levels by week 6. Descriptions of COVID-19 transmission behaviors by place are given in Table 7. Mask non-compliance was highest near non-academic campus building exits/entrances and lowest in open spaces while face touching was highest at crosswalks and lowest at campus walkway pinch points; however, mask compliance and face touching did not vary significantly by place. Conversely, not physical distancing, mask and physical distancing non-compliance, and surface touching did vary significantly by place. Individuals were less likely to physical distance in open spaces, crosswalks, and at non-node sites [*Χ*^2^ (5, N = 5914) = 17.8; *p* = 0.003]. Non-compliance with both mask wearing and physical distancing was highest at non-node sites, crosswalks and non-academic campus building exits/entrances [*Χ*^2^ (5, N = 5882) = 12.8; *p* = 0.03]. Surface touching was highest at non-academic campus building exits/entrances and academic building exits/entrances [*Χ*^2^ (5, N = 5903) = 16.1; *p* = 0.01]. A visual representation of the spatial distribution of mask and physical distancing non-compliance is provided in Figure 8.

## 4. Discussion

The purpose of this study was to create a direct observation video method (VT-scan) for assessing outcomes related to COVID-19 transmission behaviors and associated personal and environmental factors. Findings indicate that video obtained using VT-scan procedures contains information on these outcomes that can be extracted in a consistent manner. Both intra- and inter-reliability were acceptable across a heterogeneous set of behaviors, personal characteristics, and environmental settings. The utility of VT-scan was evidenced in a pilot study where we demonstrated that the distribution of COVID-19 transmission behaviors varied across personal and environmental factors as well as time. 

Direct observation, with and without the use of video, has been used extensively to study human behaviors in public open spaces (natural and man-made) [38,39,40]. A few very recent studies have used direct observation to describe mask wearing, physical distancing and personal characteristics [21,22,23,24]. Details regarding observational procedures were lacking in these studies as were more robust examinations of reliability (only inter-rater reliability was examined). Two studies restricted observations to a grocery store setting with one simply “cross-checking” data among observers and the other reporting only inter-rater reliability [21,22]. The other two investigations included outdoor observation areas, but only one of these examined the reliability of the procedure [23,24]. The results from the latter study are similar to those of the current study with 40.2% not having a mask (current study 39.2%) and 16.7% of them wearing the mask incorrectly (22.1% current study). Despite these similarities, it is difficult to compare studies given the Cohen et al. [23] employed observation methodology that has been shown to be inaccurate when describing larger groups of people [21,22,23,24].

A common finding across studies and observation approaches is that they tend to be highly reliable or consistent in discerning characteristics of interest across observers and time. The same was found in this study for extracting data on multiple COVID-19 related factors from videos of people going about their everyday business in outdoor areas. When using direct observation combined with video capture, reliability is essentially a product of data collection and data extraction. In the current study, data collection was optimized by utilizing one trained observer, a pre-determined route, and a high-resolution WVD. Thus, the videos (data) were obtained using a standardized procedure that enhanced the video review process. The effect on reliability of using alternative data collection procedures (multiple observers, other video devices, etc.) definitely warrants examination. However, it is recommended to at least pre-plan a route and use a high-resolution video device. Data extraction is dependent on the quality of the video (set during data collection), the decision rules for labeling characteristics of interest, and how well reviewers follow the rules (set by training). Our aim was to create decision rules that better aligned VT-scan with the “objective” nature of direct observation. A set of criteria were established for the outcomes to lessen the reliance of the reviewers on their own “subjective” criteria which is an advancement over other observation approaches used to study human behavior [41,42]. The results clearly show that the training effort and decision rules worked in unison (along with quality videos) to produce highly reliable evaluations of COVID-19 transmission behaviors and related factors including geographical location. 

The case study further demonstrated the value of VT-scan. Characteristics were assigned to most people seen in the videos along with when and where they were when observed. This is consistent with other studies that have repeatedly shown that the observation method is useful for providing information on humans such as age, sex/gender, race/ethnicity, and physical activity (type and intensity) as well as the behavior settings they are in [41,42]. In addition, outcomes were similar to what has been noted in other studies in the US utilizing surveys and surveillance camera data (e.g., face touching 7.7%; mask wearing 53%) [18,43]. Again, the utility of the observation method rests heavily on the rules devised to guide observers’ descriptions of the people they see and how well they are trained on these rules. This is the case whether characteristics are recorded in the field or from a video. In the current study, video reviewers were tasked with identifying a fairly large number of characteristics compared with what other observation methods require. For example, four primary characteristics of park users are identified with the SOPARC while only one is ascertained with the BWM [41,42]. This facet of VT-scan strongly supports the use of video. Although an observer could record the data in the field, it would be a very difficult task especially with larger groups and when trying to determine geographical locations. Further, observations would almost have to be done from stationary points. This would eliminate a dynamic aspect of VT-scan—a mobile observer. With VT-scan, observations were done along a 4.6 mile route which undoubtedly enhanced the diversity of settings and physical/social environments in which the observed behaviors occurred. A comprehensive picture like this is a major principle of theories explaining human behavior such as the Social Ecological model [44]. 

There are other added benefits to incorporating video into the direct observation method. First, data from video vs. in-person only assessments tend to be more accurate simply because computer software can be used to manipulate the videos (e.g., pause, zoom) increasing the odds the outcome of interest is correctly identified [21,22]. Video also allows for retrospective analysis should a variable of interest arise in the future. For instance, if new literature emerged suggesting eyeglasses may reduce the transmission of COVID-19, it would be perfectly within our ability to revisit the videos to describe this characteristic. It should also be pointed out that the time required to extract the relevant information from the videos could preclude widespread use of the method due to lack of resources (finances). This, however, may be only a temporary drawback given the exponentially growing artificial intelligence (AI) solutions being developed. Algorithms currently exist to detect mask wearing behavior in videos from stationary cameras and they could potentially be extended to include other data (e.g., physical distancing, personal characteristics) [16,17,18,19]. In addition, the VT-scan video capture protocol was constructed with the idea of eventually using AI to automatically extract relevant data. It is theoretically possible to utilize AI techniques such as computer vision to analyze video obtained using VT-scan. Once these advancements are made, time requirements for video processing will be drastically reduced resulting in financial savings. 

The VT-scan was useful for describing COVID-19-related factors as well as their interactions with each other. Further, the sensitivity of VT-scan was exemplified in the time fluctuations noted over the course of case study observations. During the case study, a State of Emergency was in effect that mandated mask wearing and physical distancing in indoor and outdoor areas with any failure to comply constituting a criminal offence. However, these rules were not enforced until week six of the case study and at that point, VT-scan noted a substantial rise in mask/physical distancing adherence. Of importance was the findings on the variation in COVID-19 transmission behaviors by personal and environmental factors. This is consistent with previous research on variations in infections and transmission behaviors across study sample characteristics [43]. However, the findings are also novel in the sense that VT-scan provides a more localized picture (smaller geographical scale) of these phenomena. In addition, VT-scan might afford a look at more nuanced deviations that arise because the data are better representative of a smaller-scale or a more defined area (e.g., campus) than a larger-scale area (e.g., state). Therefore, VT-scan could be used to enhance agent-based modeling, local policy and/or interventions or campaigns aimed at reducing viral spread. Relatedly, localized anomalies relative to place may be uncovered. For example, in the case study we found lower percentages of individuals practiced both mask wearing and physical distancing when in open spaces and along campus walkways. Armed with such information, highly “tailored” prevention/mitigation efforts could be put forth—University administration could curtail traffic on campus walkways (e.g., through staggered class times) or use “point-of-decision” prompts (e.g., signs near campus open spaces encouraging mask use and physical distancing). Such interventions have been shown effective for increasing stair vs. escalator/elevator use [45]. Additionally, VT-scan allows for flexibility in terms of where data are collected which is not a strength of other assessment methods in this area especially those relying on surveillance cameras. Theoretically, data collection could be adjusted to capture scheduled and unscheduled (pop-up) large gatherings or uncover small-scale anomalies that could factor into transmission rates. As an example of the latter, we observed low physical distancing compliance along a retail business corridor adjacent to campus. It is possible that University student COVID-19 risk/rates are enhanced because they utilize this business area, but such utilization is not accounted for in University COVID-19 prevention policies or interventions thus potentially reducing their effectiveness.

While the case study was done to demonstrate the utility of VT-scan, the links observed between demographic characteristics and transmission behaviors warrant discussion. First, individuals not moving (sitting or standing) were less compliant with masks, physical distancing, and surface touching than those moving. Most of the non-movers were sitting at tables where table dimensions did not permit physical distancing. In addition, people at tables could remove masks according to University and State mandates under the guise that they were drinking or eating (even though tables were in public areas and not directly associated with a restaurant). Whites compared with non-whites and females compared with males were less compliant with masks while not physical distancing whereas obese individuals were less likely to practice physical distancing. These findings agree with previous studies using self-report methodology and may reflect a myriad of reasons including perceived risk (e.g., whites perceive less risk of contracting and dying from COVID-19 than other racial groups resulting in less compliance), social and economic factors (e.g., those with more resources to address illness may feel less pressure to conform), and personal beliefs/attitudes (e.g., men view mask wearing as less masculine than women which deters the behavior when in public) [43,46,47,48,49]. While determining why transmission behaviors vary among population subgroups was beyond the scope of this study, it may be possible to utilize VT-scan along with other methods (e.g., field surveys) in future studies to examine this topic more closely. 

The findings of this study should be considered in the context of its limitations. Data were collected during two months in the late summer early fall on and near a mid-sized University campus perhaps restricting generalizations to dissimilar seasons/areas. However, the influence of this limitation may be somewhat diminished given sections around campus represented typical residential and business zones. Nevertheless, unforeseen issues may arise when extending VT-scan to different areas or even different situations. For example, the accuracy of direct observation has been shown to fall off slightly when larger groups of people are observed (as might be found during rush hour in New York City) [40]. Relatedly, VT-scan was examined during midday and it is possible that its accuracy could be affected by phenomenon associated with time of day such as shadowing. A previous study did not find this to be a problem while using observation to describe different types of physical activities occurring on sidewalks/streets [25]. Nevertheless, more nuanced behaviors like masking might be more difficult to discern under different lighting conditions. While the reliability of VT-scan was the main metric examined, validity is also important especially when describing personal characteristics such as age. Although there is some work validating physical activity intensity (moderate vs. vigorous exertion) estimates using direct observation, there is nothing on the validity of direct observation (with or without video) for the specific characteristics examined in this study [41].

## 5. Conclusions

This study provides evidence on the reliability of a new video-enhanced, direct observation method (VT-scan) for systematically assessing COVID-19 transmission behaviors and related factors. It demonstrates that data of this nature can be successfully captured in and extracted from video. Although VT-scan’s utility for wide-spread, surveillance efforts may be limited and only reach fruition when integrated with AI techniques, the method is highly replicable and has immediate value for informing localized, agent-based models and local policy aimed at mitigating the COVID-19 pandemic as well as other and future epidemics and pandemics due to airborne pathogens.

## Figures and Tables

**Figure 1 ijerph-18-09329-f001:**
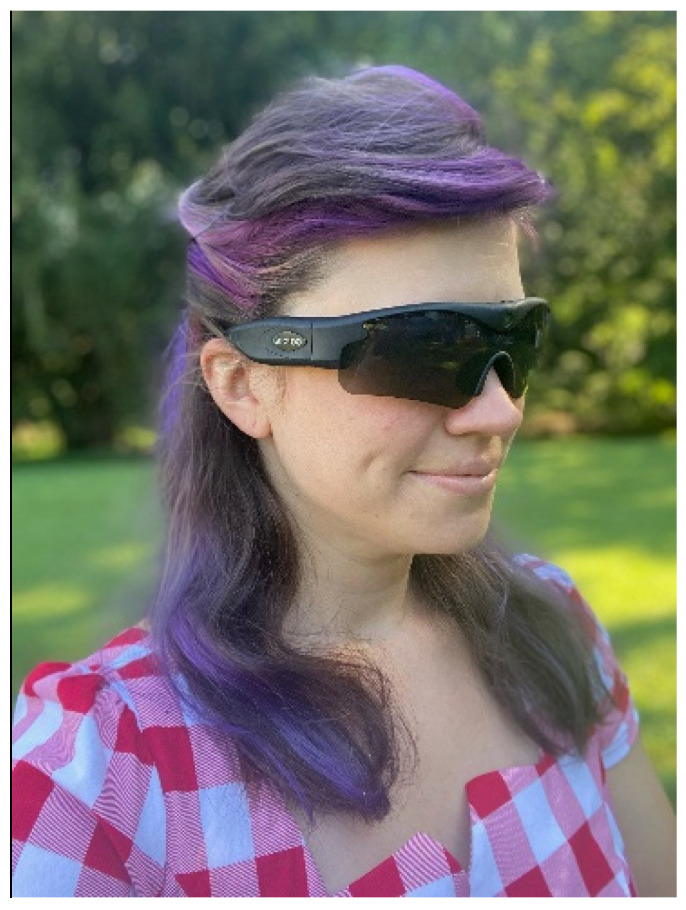
Gogloo E7 SMART WVD. Written informed consent was obtained from the individual for the publication of this image.

**Figure 2 ijerph-18-09329-f002:**
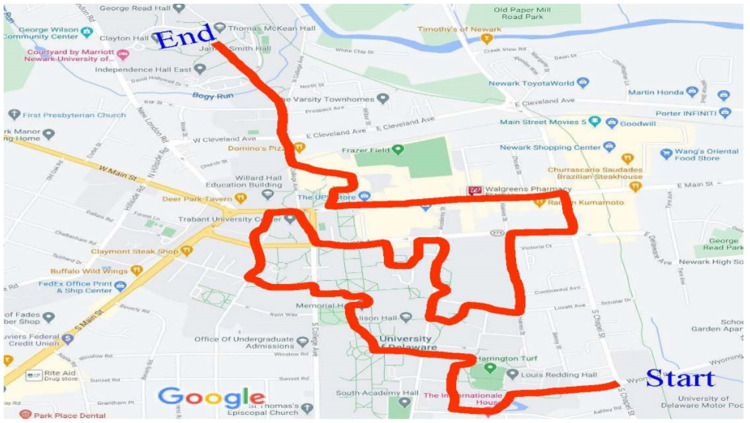
Map of route.

**Figure 3 ijerph-18-09329-f003:**
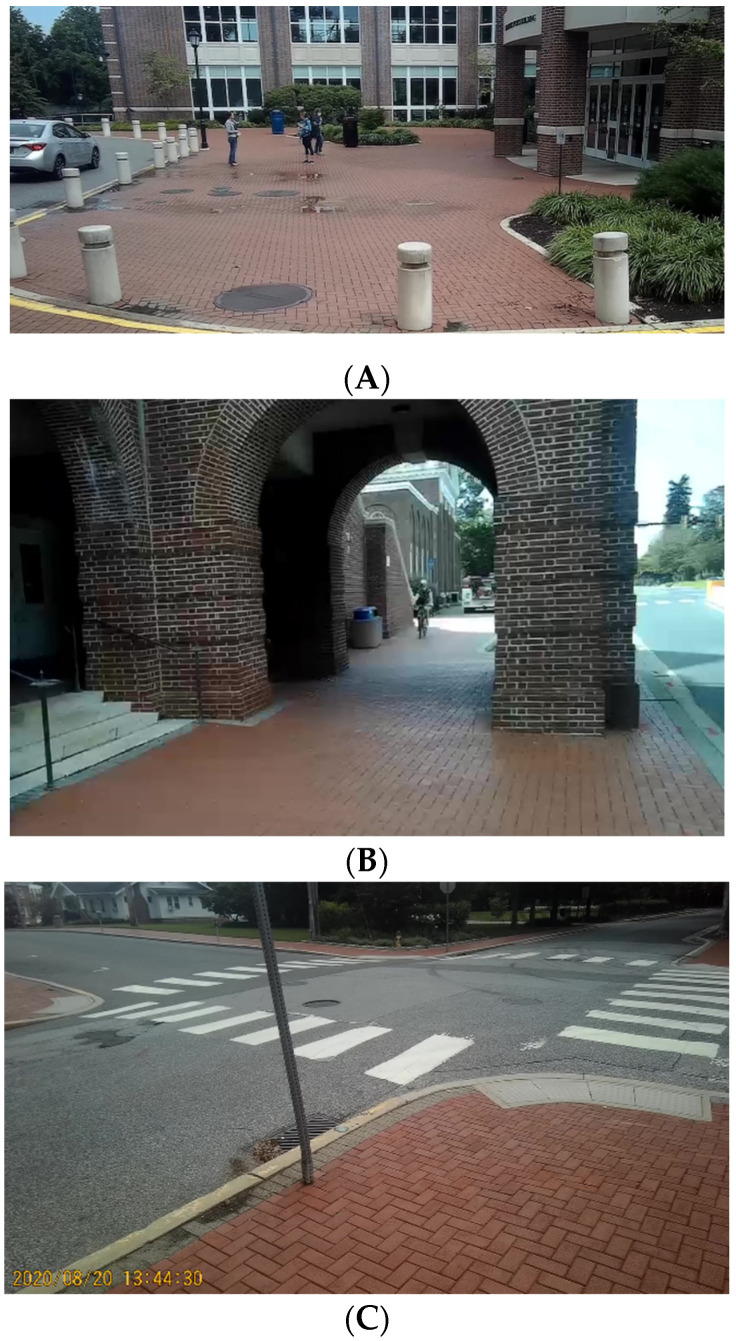
Examples of nodes (**A**): entrances/exits to campus buildings; (**B**): narrow walkway; (**C**): crosswalks; (**D**)—public open spaces.

**Figure 4 ijerph-18-09329-f004:**
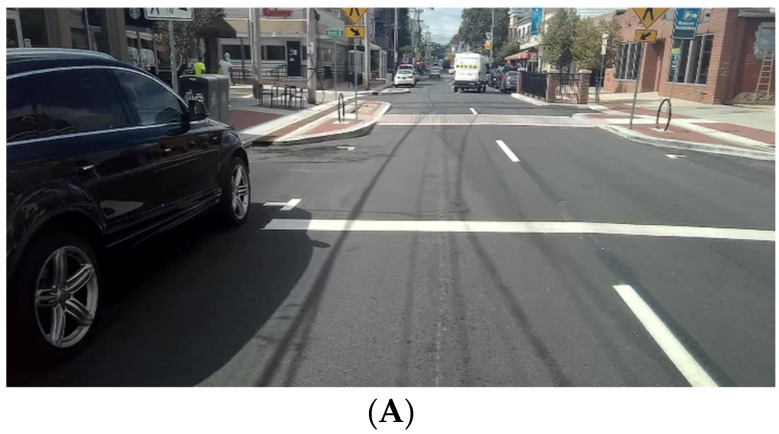
Street view with facial capture (**A**): people orientated in same direction as observer; (**B**): side video of people seen in (A).

**Figure 5 ijerph-18-09329-f005:**
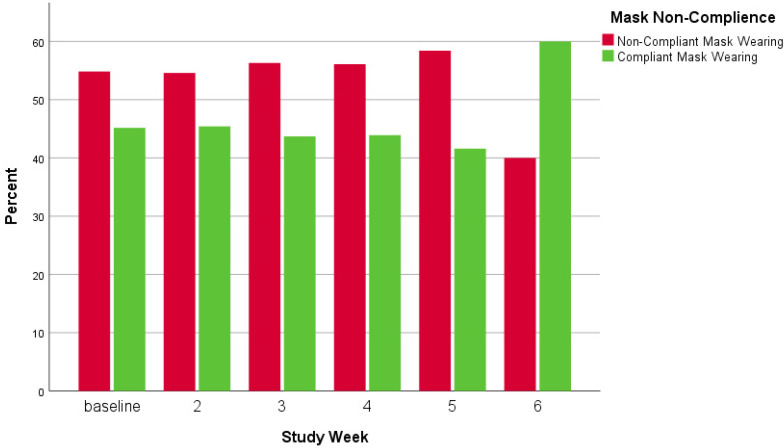
Mask wearing compliance by study week.

**Figure 6 ijerph-18-09329-f006:**
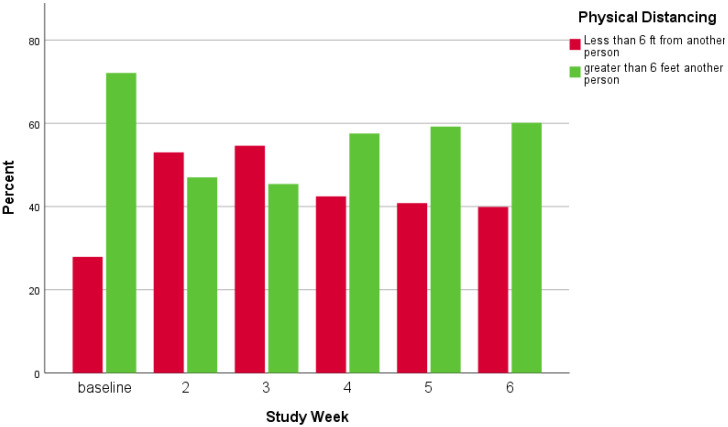
Physical distancing by study week.

**Figure 7 ijerph-18-09329-f007:**
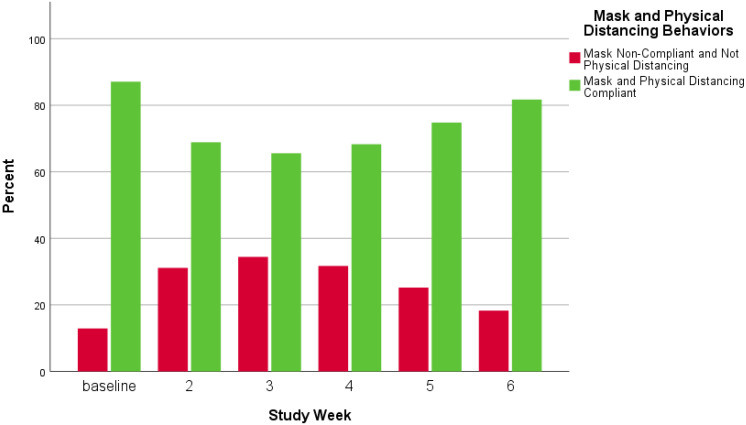
Mask and physical distancing behaviors combined by study week.

**Figure 8 ijerph-18-09329-f008:**
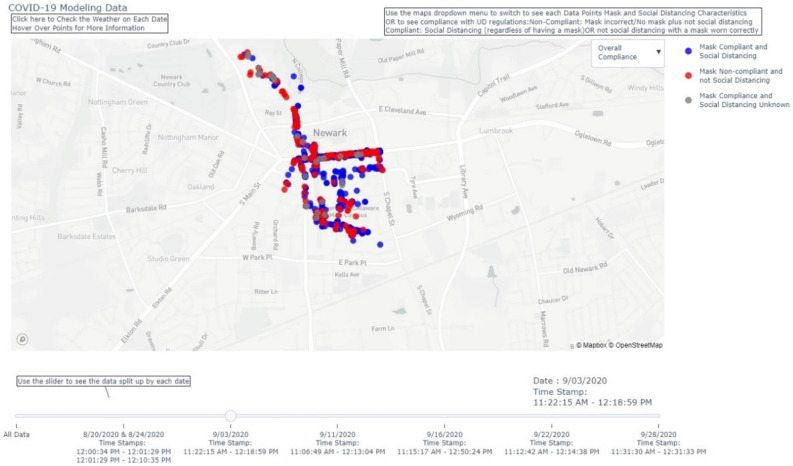
Spatial distribution of mask and physical distancing non-compliance.

**Table 1 ijerph-18-09329-t001:** Rules used for training and obtaining outcomes from videos.

General Nodes: Attempt to describe all people at nodes. Inter-node: Attempt to describe all people who are passed by the observer when: - On campus walkways that are part of the inter-node route. - On sidewalks/streets that are part of the inter-node route. - Exclude people not in these areas (e.g., in a parking lot next to a campus walkway). When describing a group of people, start description with person farthest left. Exclude patrons in outdoor seating areas of bars/restaurants. Seating areas are clearly demarcated by a fence. Apply timestamp when first characteristics is recorded for a particular person. Do not include people in vehicles or inside buildings/stores/etc.
Personal Characteristics Activity People sitting/standing labeled as “not moving”. All others are labeled “moving” including those walking, running, biking, other. If a non-mover begins moving while still in the video, label them as a mover. Sex Sex traits examined: Body size/stature (males larger), facial features, enlargement of breasts. Gender traits examined: clothes, hair style. Age Group Characteristics examined: body size/stature (shorter, smaller = <18 years), facial features, hair color, clothes. Race Characteristics examined: skin tone, hair features, facial features [29]. Weight Status Not obese vs. obese according to adult and child obesity scales [30,31].
COVID-19 Transmission Behaviors Have a mask Visible face covering/mask anywhere on person. Mask worn correctly Covers nose and mouth. No ventilation area. Physical distancing Compliance <6 feet from another person. Non-compliance >6 feet from another person. Distance between people determined using: - Google maps measurement tool along with long., lat. of relevant people - Google maps measurement tool along with a standard (objects of known length in the video—e.g., sidewalk width) - Arms length—about 2 arm lengths apart = compliance - Person is alone = compliance Touched fomite Any hand to fomite interaction Fomite defined as any inanimate object Touched fomite Any hand to fomite interaction

**Table 2 ijerph-18-09329-t002:** Intra- and inter-class correlation coefficients for each outcome measure.

Intra-Rater Reliability		
Outcome	Number Described (%) ^a^	ICC ^b^ (*p* Value)
Total Described R1 ^c^ Total Described R1 ^d^	126 129	0.993 (<0.001)
Activity (moving) R1 Activity (moving) R1	96 (76.2) 97 (75.2)	0.935 (0.005)
With mask R1 With mask R1	84 (66.7) 88 (68.2)	0.858 (0.014)
Mask correct R1 Mask correct R1	102 (81.0) 105 (81.4)	0.906 (0.010)
No Physical Distance R1 No Physical Distance R1	67 (53.2) 68 (52.7)	0.935 (0.005)
Did not touch surface R1 Did not touch surface R1	108 (85.7) 105 (81.4)	0.997 (<0.001)
No face touching R1 No face touching R1	121 (96.0) 115 (89.1)	0.991 (<0.001)
Age Group 18-30y R1 Age Group 18-30y R1	90 (71.4) 90 (69.8)	0.881 (0.015)
White R1 White R1	110 (87.3) 104 (80.6)	0.993 (<0.001)
Female R1 Female R1	73 (57.9) 73 (56.6)	0.979 (<0.001)
Not Obese R1 Not Obese R1	95 (75.4) 98 (76.0)	0.993 (<0.001)
Inter-Rater Reliability		
Total Described R1 ^b^ Total Described R2 ^e^	126 131	0.996 (<0.001)
Activity (moving) R1 Activity (moving) R2	96 (76.2) 102 (77.9)	0.905 (0.007)
With mask R1 With mask R2	84 (66.7) 90 (68.7)	0.971 (0.001)
Mask correct R1 Mask correct R2	102 (81.0) 107 (81.7)	0.970 (0.001)
No Physical Distance R1 No Physical Distance R2	67 (53.2) 66 (50.4)	0.925 (0.004)
Did not touch surface R1 Did not touch surface R2	108 (85.7) 100 (76.3)	0.920 (0.005)
No face touching R1 No face touching R2	121 (96.0) 120 (91.6)	0.987 (<0.001)
Age Group 18-30y R1 Age Group 18-30y R2	90 (71.4) 93 (71.0)	0.836 (0.019)
White R1 White R2	110 (87.3) 116 (88.5)	0.976 (<0.001)
Female R1 Female R2	73 (57.9) 72 (55.0)	0.870 (0.012)
Not Obese R1 Not Obese R2	95 (75.4) 111 (84.7)	0.872 (0.012)

^a^ (% of total for outcome per reviewer and review session); ^b^ ICC—intraclass correlation coefficient; ^c^ R1—reviewer 1, review session 1; ^d^ R1—reviewer 1, review session 2; ^e^ R2—reviewer 2, review session 1.

**Table 3 ijerph-18-09329-t003:** Intra- and inter-reviewer reliability for presence/absence of characteristics.

	Intra (*n* = 111)		Inter (*n* = 117)	
Outcome	* κ *	% Agreement	* κ *	% Agreement
Activity	0.877	96.7	1.00	100
Mask	0.724	89.0	0.732	89.7
Mask Compliance	0.875	95.0	0.898	96.2
Physical Distancing	0.931	96.7	0.949	97.4
Touch surface	0.735	87.9	0.681	87.2
Touch face	0.775	95.6	0.875	97.5
Age group	0.812	92.3	0.703	84.6
White	0.729	89.3	0.713	89.7
Sex	0.810	91.2	0.794	89.7
Obese	0.843	94.5	0.776	92.2

*κ*—Kappa Statistic; All *κ* statistics are significant at *p* < 0.001.

**Table 4 ijerph-18-09329-t004:** Latitude and longitude coordinate comparisons.

	Intra (*n* = 111)		Inter (*n* = 117)	
Outcome	Mean (SD)	*p* Value	Mean (SD)	*p* Value
Not moving Moving	5.52 (3.57) 3.06 (3.11)	0.17	1.77 (1.47 3.61 (3.25)	0.14
Not Physical Distancing Physical Distancing	4.45 (4.05) 3.06 (2.63)	0.31	2.71 (2.64) 3.29 (3.18)	0.62
Node Non-Node	3.13 (2.79) 3.40 (3.66)	0.52	1.89 (3.72) 3.56 (2.44)	0.18

**Table 5 ijerph-18-09329-t005:** Subject characteristics relevant to COVID-19 transmission behaviors (*n* = 916 observed).

Characteristics and Behaviors	Percent of # Described for That Outcome
<18 years (*n* = 13)	1.4
18–30 years (*n* = 653)	72.5
31–55 years (*n* = 188)	20.9
>55 years (*n* = 46)	5.1
(Could not be determined *n* = 16)	
Non-White (*n* = 170)	18.9
White (*n* = 728)	81.1
(Could not be determined *n* = 18)	
Male (*n* = 413)	45.4
Female (*n* = 497)	54.6
(Could not be determined *n* = 6)	
Not obese (*n* = 728)	80.7
Obese (*n* = 174)	19.3
(Could not be determined *n* = 14)	
Not moving (*n* = 262)	28.6
Moving (*n* = 654)	71.4
(Could not be determined *n* = 0)	
No mask (*n* = 350)	39.1
With mask (*n* = 544)	60.9
(Could not be determined *n* = 22)	
Incorrect mask use for those with mask (*n* = 118)	22.1
Correct mask use for those with mask (*n* = 418)	87.9
(Could not be determined *n* = 8)	
Mask non-compliance (*n* = 468)	52.8
Mask compliance (*n* = 418)	47.2
(Could not be determined *n* = 30)	
Consequences of mask non-compliance	
Nose exposed (*n* = 61)	13.1
Nose and mouth exposed (*n* = 407)	86.9
(Could not be determined *n* = 0)	
Not physical distancing <6 ft (*n* = 415)	45.4
Physical distancing >6 ft (*n* = 499)	54.5
(Could not be determined *n* = 2)	
Non-compliant with mask & physical distancing (244)	27.6
Compliant with mask & physical distancing (640)	72.4
(Could not be determined *n* = 32)	
Did not touch face (*n* = 843)	93.3
Touched face (*n* = 61)	6.7
(Could not be determined *n* = 12)	
Did not touch surface (*n* = 737)	81.5
Touched surface (*n* = 167)	18.5
(Could not be determined *n* = 12)	
Surfaces touched	
Cell Phone (*n* = 64)	40.8
Auto, building, railing, door handle (*n* = 39)	24.8
Bench (*n* = 22)	14
Table (*n* = 18)	11.5
Trash can, parking meter, (*n* = 7)	4.5
Tools (*n* = 7)	4.5
(Could not be determined *n* = 10)	

**Table 6 ijerph-18-09329-t006:** COVID-19 transmission behaviors by demographics.

Characteristic	Mask Non-Compliance	Not Physical Distancing	Mask and Physical Distancing Non-Compliant	Touched Surface	Touched Face
18–30 years	51.2	46.4	27.1	17.2	6.9
31–55 years	56.1	41.3	26.6	23.2	6.2
>55 years	47.6	38.6	21.4	20.9	4.7
Non-White	31.8	37.5	13	18.7	7.1
White	57.5	47.7	30.9	18.8	6.7
Male	52.4	36.4	23.4	18	5.4
Female	53.1	53.1	31.1	18.9	8
Not obese	53.8	43.1	27	17.6	6
Obese	47.2	54.5	29.2	21.2	9.7
Not moving	65.4	53	39.4	42.3	6.9
Moving	47.6	42.5	22.9	8.8	6.7

**Table 7 ijerph-18-09329-t007:** COVID-19 transmission behaviors by specific place of observation.

Area	Mask Non-Compliance	Not Physical Distancing	Mask and PD Non-Compliant	Touched Surface	Touched Face
Crosswalks	46.1	49.4	29.1	15.9	11.2
Non-academic campus building exits/entrances	64.3	23.3	26.9	43.8	6.3
Open spaces	36.1	50.0	16.7	11.1	2.8
Academic building exits/entrances	50.0	15.4	7.7	23.1	3.8
Campus walkway pinchpoints	51.1	40.8	14.9	17.4	2.3
Internode areas	55.5	47.0	30.0	18.2	6.2

## Data Availability

The data presented in this study are available on request from the corresponding author. The data are not publicly available due to privacy restrictions.

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
