# Peer review of "A Direct Observation Video Method for Describing COVID-19 Transmission Factors on a Micro-Geographical Scale: Viral Transmission (VT)-Scan"

_ijerph, 2021, doi:10.3390/ijerph18179329_

Round 1

Reviewer 1 Report

  1. It would be good to talk about the official mask mandate in that area during the study. That would also have an impact on compliance and needs to be discussed.
  2. The drawback of the study is that the current manual methods described to detect and analyze might not be easily scalable and repeatable. The authors do talk about “growing artificial intelligence solutions being developed.“ Can the current video recordings be used for AI & ML modeling and analysis of the data? Please discuss future study plans around that if any.
  3. Need reference or documentation for “The University’s Institutional Review Board for Protection of Human Subjects Committee waived the requirement for writ-ten informed consent for participants in this study under exemption #2: Research involving observation of public behavior, in accordance with the national legislation and the institutional requirements.”
  4. The accuracy of direct observations falls off when larger groups are observed. How about the difference between day and night? Are these measurements going to be as accurate during the right? Please discuss.

Reviewer 2 Report

In this article, the authors present a modern method of surveillance allowing a detailed analysis of the behavior of populations in order to provide reliable data to political decision-makers.

The description of the method is quite clear, although its use can be difficult to apply.

Some limitations of this study are mentioned by the authors in the discussion, such as the limited period of data collection. It would be interesting to collect this data over a period of at least a year to draw conclusions.

Also, no virologic characteristics are considered. The association of a specialist in virology could bring another point of view and draw more interesting conclusions on the consequences of behavior in the transmission of respiratory viruses.

It is not very clear how this study provides information related to SARS-CoV-2 or any virus of respiratory transmission.

Would it be possible to specify what “the H1N2 Coronavirus” corresponds to? Is this an error that has crept into the abstract following a mix between the H1N2 influenza virus and the coronaviruses?

Reviewer 3 Report

This paper presents a new method to understand better COVID-19 transmission: via a direct observation video method to analyze behavioral patterns of compliance or non-compliance to prevention rules (mask wearing & social distancing).

Though the validity of this method of analysis is convincing as well as its implementation appears to respect scientific protocols, there are still some issues with the paper that could be addressed.

The main problem with the paper is that it's not really discussing its results.

The "Table 6. COVID-19 transmission behaviors by demographics" presents to the reader interesting figures; however there is never an in-depth discussion and analysis of these results.

While the summary said: "Transmission behaviors varied by demographics with white, obese males least likely to be mask-compliant and white, overweight/obese females least likely to physical distance." There is almost nothing in the paper related to a closer analysis of these differences. 

While it is said in the introduction "It should be pointed out, however, that while masks are efficacious their effectiveness is questionable mainly due to adherence resistance resulting from a myriad of reasons such as the politicization of mask use" There is nothing in the rest of the paper that could help us to substantiate this claim or to know whether the paper adds something new or not to the discussion about "politicization of mask use".

More precisely: why there is this difference between "Normal" & "Overweight/obese" in Mask compliance 53.8 / 47.2 and Social Distancing 43.1/54.5?  Many questions here: 1. why using this criteria "Normal" & "Overweight/obese" and not any other one? 2. What is the definition of overweight & obese used by the authors? 3. How can they assess it without directly weighing the people seen in the videos? Moreover: 4. How to understand this difference? What to make about it? 5. How to understand also that mask wearing compliance is lower for "Overweight/obese" but social distancing better? 6. What are the social, cultural, political factors behind this difference?

The same kind of questioning hold for the most striking difference: between "White" & "non-White" in Mask compliance 31.8 / 57.5  and Distancing 37.5/47.7. Many questions again: 1. what is meant by "non-White" here: not very clear and rather vague? 2. How to justify to use this criteria? 3. How to understand that both mask wearing compliance & Social Distancing is lower for "White" people? 4. Why the difference in mask wearing compliance is so strong between "White" & "non-White"? 5: What are the social, political and cultural factors conditioning such a difference in behaviors?

Moreover, how to make any use of these data if we don't know if they have any statistical meaning? If we don't know the statistical proportion of "Normal" & "Overweight/obese" normally transiting in this area or usually populating it, it seems difficult to make any meaningful conclusion or generalization about difference in terms of mask compliance & social distancing. Even more for the statistical proportion of "White" & "non-White" people: is the difference between White & non-White due to difference in behavioral pattern or does it simply reflect the demography of the area? To provide a picture of the global demographic of the area will be helpful to assess the statistical value of the differences observed in terms of behavior patterns related to different category of people.

It can be also added that while the paper stresses that mask compliance are lower for White it provides a picture with a non-compliant mask wearing person belonging to the "non-White" category (figure 4).

Finally, in the introduction perhaps the presentation of previous research is not developed enough. Difficult to understand what Susceptible-Infected-Recovered (SIR) really means. 

And also "In just over a single year, there have been 191 million cases globally with over 4 million virus-related deaths" Today cases are calculated to be 204 million - so these figures will need to be updated. 

Round 2

Reviewer 2 Report

The article was ameliorate

Reviewer 3 Report

The authors of the paper have taken into account very consciously and professionnally the remarks we previously raised by adding new and relevant information as well as addressed the problems we mentioned in a meaningful way. As far as we are concerned, the paper can be accepted in its current form.